

# The maintenance of elevated active chlorine levels in the Antarctic lower stratosphere through HCl null-cycles

Rolf Müller[1], Jens-Uwe Grooß[1], Abdul Mannan Zafar[1], and Ralph Lehmann[2]

[1]Institute of Energy and Climate Research (IEK-7), Forschungszentrum Jülich, 52425 Jülich, Germany
[2]Alfred Wegener Institute, Helmholtz Centre for Polar and Marine Research, Potsdam, Germany

*Correspondence to:* R. Müller (ro.mueller@fz-juelich.de)

**Abstract.** The Antarctic ozone hole arises from ozone destruction driven by elevated levels of ozone destroying ("active") chlorine in Antarctic spring. These elevated levels of active chlorine have to be formed first and then maintained throughout the period of ozone destruction. It is a matter of debate, how this maintenance of active chlorine is brought about in Antarctic spring,

when the rate of formation of HCl (considered to be the main chlorine deactivation mechanism in Antarctica) is extremely high. Here we show that in the heart of the ozone hole (16-18 km or 100-70 hPa, in the core of the vortex), high levels of active chlorine are maintained by effective chemical cycles (referred to as HCl null-cycles hereafter). In these cycles, the formation of HCl is balanced by immediate reactivation, i.e. by immediate reformation of active chlorine. Under these conditions, polar stratospheric clouds sequester $HNO_3$ and thereby cause $NO_2$ concentrations to be low. These HCl null-cycles allow active

chlorine levels to be maintained in the Antarctic lower stratosphere and thus rapid ozone destruction to occur. For the observed almost complete activation of stratospheric chlorine in the lower stratosphere, the heterogeneous reaction $HCl + HOCl$, the production of HOCl via $HO_2 + ClO$, with the $HO_2$ resulting from $CH_2O$ photolysis, is essential. These results are important for assessing the impact of changes of the future stratospheric composition on the recovery of the ozone hole. Our simulations indicate that, in the lower stratosphere, future increased methane concentrations will not lead to enhanced chlorine deactivation

(through the reaction $CH_4 + Cl \longrightarrow HCl + CH_3$) and that extreme ozone destruction to levels below $\approx 0.1$ ppm will occur until mid-century.

## 1 Introduction

Because of the success of the Montreal protocol and its amendments and adjustments, the atmospheric halogen loading peaked in the early nineties and has been declining slowly since then. However, even today, the Montreal Protocol has already achieved

significant benefits for the ozone layer and the Antarctic ozone hole (by 2013, Chipperfield et al., 2015). Nonetheless, model simulations predict that the Antarctic ozone hole will continue to occur for decades (e.g., WMO, 2014; Oman et al., 2016; Fernandez et al., 2017).

The rapid chemical destruction of ozone in the Antarctic stratosphere in spring is caused by catalytic cycles driven by ClO and BrO (McElroy et al., 1986; Molina and Molina, 1987; Solomon, 1999). To run efficiently, these catalytic cycles require

large concentrations of "active chlorine, $ClO_x$" ($ClO_x = ClO + 2 \times Cl_2O_2 + Cl$). The activation of chlorine, i.e., the conversion



of the main chlorine reservoir species (HCl and ClONO$_2$) to ClO$_x$ occurs through heterogeneous reactions (Solomon, 1999). The initial step of chlorine activation proceeds via the heterogeneous reaction (Solomon et al., 1986)

$$ClONO_2 + HCl \rightarrow Cl_2 + HNO_3 \quad ; \tag{R1}$$

this step occurs very rapidly during polar night in about mid-May when temperatures become low enough for heterogeneous chlorine activation. Because of the initial concentration of HCl (before the onset of heterogeneous reactions) in the polar vortex being greater than that of ClONO$_2$ (Jaeglé et al., 1997; Santee et al., 2008), the amount of Cl$_2$ produced initially in reaction R1 is limited by the amount of available ClONO$_2$ (Salawitch et al., 1988; Crutzen et al., 1992; Portmann et al., 1996). The further chemical activation to near zero HCl values, as observed in Antarctic winter and in cold winters in the Arctic (Jaeglé et al., 1997; Santee et al., 2005; Santee et al., 2008; Manney et al., 2011; Wegner et al., 2012) requires the reproduction of partners for heterogeneous reactions with HCl.

With the return of sunlight to the polar region a period follows, characterised by further activation and maintenance of high levels of active chlorine (as observed, Santee et al., 2005; Santee et al., 2008) during which most of the ozone depletion occurs. Polar stratospheric clouds are measured in the Antarctic lower stratosphere until early October (Pitts et al., 2009). However, which chemical processes are responsible for further activation and maintenance of active chlorine is a matter of debate (Solomon et al., 2015).

Here, for the heart of the ozone layer (in the core of the vortex, in the lower stratosphere), where minimum ozone mixing ratios are reached (Solomon et al., 2005), we suggest the following picture of Antarctic ozone depletion. First, as the initial step of chlorine activation, the available ClONO$_2$ is titrated against HCl via Reaction R1. Then, very little chemical change occurs in polar night until early August ("sleeping chemistry") and a relatively slow additional chlorine activation until early September. The maintenance of high ClO$_x$ values during mid-September to early October is accomplished by effective reaction cycles ("HCl null-cycles") which chemically inhibit a deactivation of chlorine that would otherwise proceed via net HCl formation. The period mid-September to early October is the period during which most of the Antarctic ozone loss occurs. This period of high ClO$_x$ ends abruptly with the rapid formation of HCl leading to deactivation of chlorine (Grooß et al., 1997; Grooß et al., 2011).

We present box-model calculations of Antarctic chlorine chemistry and ozone depletion which allow the chemical mechanisms and the impact of particular reactions to be studied in detail. In particular, we apply a detailed analysis of the temporal development of the rates of the key chemical processes based on a unique algorithm for the determination of chemical pathways (Lehmann, 2004). We demonstrate that for the efficacy of the HCl null-cycles it is essential that the heterogeneous reaction (Prather, 1992; Crutzen et al., 1992)

$$HOCl + HCl \rightarrow Cl_2 + H_2O \tag{R2}$$

as well as the gas-phase reaction (Crutzen et al., 1992)

$$CH_3O_2 + ClO \rightarrow CH_3O + ClOO \tag{R3}$$



(where the product ClOO decomposes rapidly to Cl and $O_2$) occurs. Further, the related formation of $HO_x$ radicals from $CH_3O$ photolysis is important.

## 2 Methods

### 2.1 Model description

The simulations presented here were performed with the Chemical Lagrangian model of the Stratosphere (McKenna et al., 2002; Grooß et al., 2005) (CLaMS); the set-up follows closely one used earlier (Grooß et al., 2011; Zafar, 2016). The model is used in box-model mode, where stratospheric chemistry is calculated for air parcels along three-dimensional trajectories. The air parcels are defined by the location and time of minimum ozone soundings in the ozone hole period, from which trajectories are calculated both backward to June and forward to December. In this way, trajectories from the core of the vortex in the lower

stratosphere are selected. The trajectories of the air parcels were calculated using wind and temperature data from operational analyses from the European Centre for Medium-range Weather Forecasts (ECMWF). The latitudinal range covered by the trajectory in June, July and August is roughly 60°S to 80°S, in September roughly 70°S to 90°S, and in October and November roughly 60°S to 85°S. The diabatic descent rates were calculated using a radiation code (Grooß et al., 2011).

The initial values for the main trace gases at the start of the simulation (1 June 2003) are: $O_3$ = 2.2 ppm, $H_2O$ = 4.1 ppm,

$CH_4$ = 1.2 ppm, $HNO_3$ = 4.5 ppb, HCl = 1.05 ppb, $ClO_x$ = 1.01 ppb, $ClONO_2$ = 12 ppt, HOCl = 5 ppt, $Br_y$ = 17 ppt, CO = 16 ppb. The sensitivity of the results of the simulations on the initial ozone mixing ratio is discussed in the appendix. (All these values are given in molar mixing ratio).

In Antarctic winter, temperatures typically fall below the threshold for chlorine activation and PSC occurrence approximately in mid-May (Pitts et al., 2009). Consistently, the initial values for chlorine species assumed here imply that the initial titration

of HCl and $ClONO_2$ (Reaction R1) has already occurred by 1 June. Likewise, the initial value assumed for $HNO_3$ implies that denitrification through the sedimentation of large NAT (nitric acid trihydrate) particles (Fahey et al., 2001; Molleker et al., 2014) had occurred by this time. This assumption likely constitutes a slightly too early onset of denitrification, but the impact of this assumption is minimal during polar night. The impact of denitrification has been explored in sensitivity studies (see appendix).

To integrate the system of stiff ordinary differential equations describing the chemistry we employ the solver SVODE (Brown et al., 1989) that does not use the family approximation. The chemical kinetic data are taken from Sander et al. (2011). The photolysis rates are calculated in spherical geometry (Becker et al., 2000) for every hour using a climatological ozone profile for ozone hole conditions from HALOE measurements (Grooß and Russell, 2005).

Of particular importance for the simulations discussed here is the representation of the photolysis of $CH_2O$. We have em-

ployed the recommended setup for both cross sections and quantum yields for 223 K (Sander et al., 2011). Using the photolysis quantum yields suggested by Röth and Ehhalt (2015) yields very similar results to those presented here (Zafar, 2016). Branching ratios of 0 and 100% cannot not occur in reality (Röth and Ehhalt, 2015) but were used here as lower and upper limits. A possible temperature dependence of both the cross sections the quantum yields of the photolysis of $CH_2O$ could be important,



as it could potentially lead to greater production of $HO_2$ in the photolysis of $CH_2O$ for the temperature range relevant here. However, an accurate temperature dependence of these kinetic data for temperatures in the polar lower stratosphere could not be considered here due to lack of laboratory information (Smith et al., 2006; Röth and Ehhalt, 2015).

Heterogeneous chemistry is calculated on ice, nitric acid trihydrate (NAT), liquid ternary particles ($H_2O/H_2SO_4/HNO_3$) and
cold liquid binary aerosols. Temperature dependent uptake coefficients of heterogeneous reactions on liquid ternary and binary aerosols are taken from parametrisations by Shi et al. (2001) as recommended (Sander et al., 2011). For uptake coefficients for reactions on NAT the parametrisation of Carslaw and Peter (1997) is used, based on laboratory measurements of Hanson and Ravishankara (1993). NAT particles are assumed to form from supercooled ternary solutions (STS) droplets. A $HNO_3$ supersaturation of three (corresponding to about $10\,K$ supercooling) is required for NAT formation. The NAT particle density
is assumed to be $3 \cdot 10^{-3}\,cm^{-3}$. This NAT particle density is lower than assumed by Grooß et al. (2011) but is in better agreement with observations.

## 2.2 Pathway analysis

A pathway in a chemical reaction system is a set of reactions converting some reactants of interest into some products of interest through some intermediate species, for which no net production or destruction occurs. An integer factor ("multiplicity") may
be assigned to each reaction. The algorithm used here for the automatic determination of all significant pathways in a chemical reaction system was developed by Lehmann (2004). As input it requires a set of chemical reaction equations and reaction rates, which are usually provided by a chemical model. Starting from the individual reactions (and their rates) as initial pathways, longer pathways are constructed step by step by connecting shorter ones. If a newly formed pathway contains sub-pathways, it is split into these. A rate for each pathway is calculated. Pathways with rates below a pre-described threshold are deleted
already during the construction process, in order to avoid an intractably large number of pathways ("combinatorial explosion").

## 3 Results

### 3.1 Maintenance of chlorine activation

We conducted box-model simulations for which the impact of mixing is neglected; however mixing across the Antarctic vortex edge is frequently overestimated substantially in current chemistry climate models (Hoppe et al., 2014). For the box-model,
the development of temperature, potential temperature and solar zenith angle along a typical air parcel trajectory in Antarctic spring is shown in Figure 1 (panels a-c). The initial titration between HCl and $ClONO_2$ via Reaction R1 occurred before June (as usually in the Antarctic, e.g. Santee et al., 2008) so that the model simulation starts with about one ppb of $ClO_x$ and with near zero values of $ClONO_2$. Little further chemical change occurs ("sleeping chemistry") as long as solar zenith angles are large (until end of July), but with decreasing solar zenith angle, HCl further decreases, leading to increasing $ClO_x$ and,
subsequently, chemical ozone destruction (e.g., Salawitch et al., 1988; Santee et al., 2005; Santee et al., 2008; Grooß et al.,



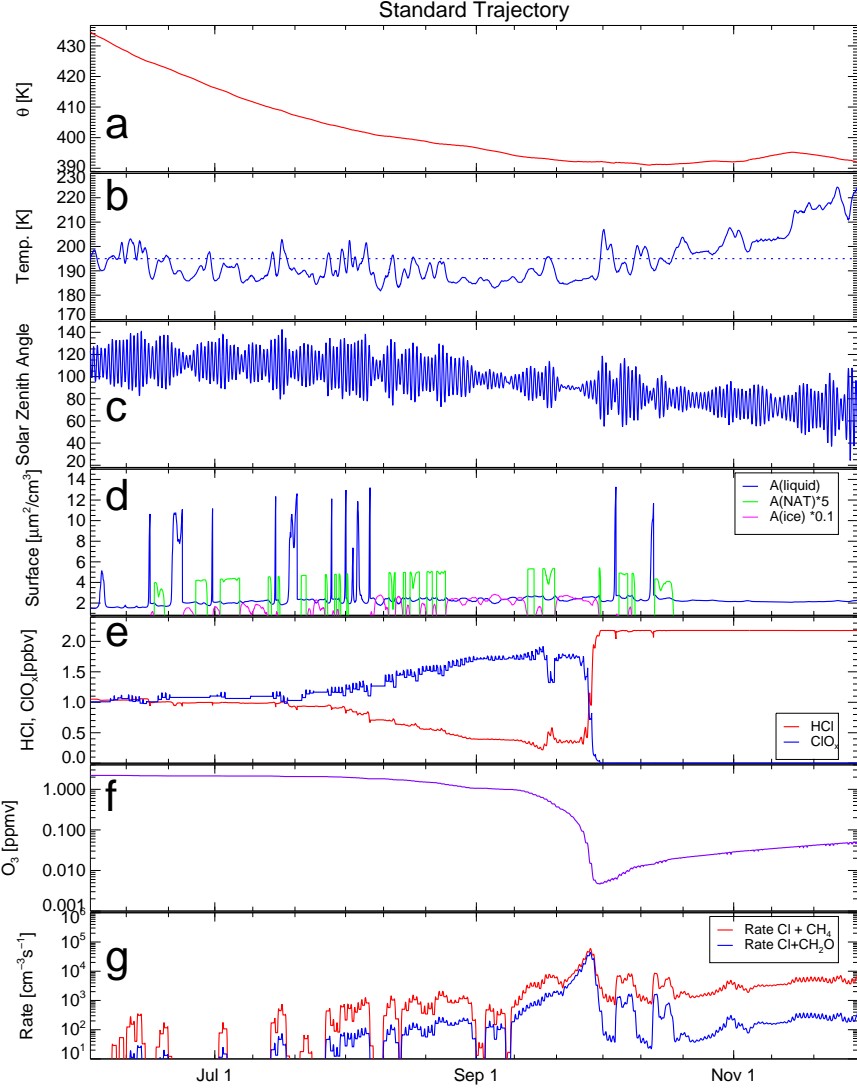

**Figure 1.** Box-model simulations along a trajectory passing through the location of the ozone sonde observation at South pole of 14 ppbv on 74 hPa (391 K) on 24 September 2003 (Grooß et al., 2011). The different panels show a time series of the relevant parameters: (a) potential temperature of the air parcel, (b) temperature, (c) solar zenith angle, (d) surface area density of ice (red, scaled by 0.1), NAT (green, scaled by 5) and liquid aerosol particles (blue), (e) $ClO_x$ (red), and HCl (blue), (f) ozone, and (g) rates of reaction of atomic chlorine with $CH_4$ (blue) and $CH_2O$ (red). For reasons of clarity, the reaction rates in panel (g) are plotted as 24 h running averages. The model simulation covers the time period from 1 June to 30 November 2003.





2011). Ozone decreases most rapidly in September and reaches minimum values of less than 20 ppb on 24 September (for the specific case considered here).

With decreasing ozone, the Cl/ClO ratio shifts increasingly towards Cl (Douglass et al., 1995; Grooß et al., 1997), so that HCl production via the reactions of Cl with $CH_4$ and $CH_2O$ (reactions R4 and R11) increases substantially by more than an order of magnitude between end of August and end of September (Figure 1, panel g). Indeed, HCl formation through reaction R4 is commonly considered as the main chlorine deactivation mechanism in Antarctic spring.

The formation rate of HCl via reactions R4 and R11 at the end of September amounts to more than 0.5 ppb per day. If these reactions proceeded unbalanced, chlorine activation and thus ozone depletion would be stopped within days. However, using an algorithm for the determination of chemical pathways (Lehmann, 2004) cycles C1 and C2 are identified for this period:

$$CH_4 + Cl \quad \rightarrow \quad HCl + CH_3 \tag{R4}$$
$$CH_3 + O_2 \quad \rightarrow \quad CH_3O_2 + M \tag{R5}$$
$$CH_3O_2 + ClO \quad \rightarrow \quad CH_3O + Cl + O_2 \tag{R3}$$
$$CH_3O + O_2 \quad \rightarrow \quad HO_2 + CH_2O \tag{R6}$$
$$ClO + HO_2 \quad \rightarrow \quad HOCl + O_2 \tag{R7}$$
$$HOCl + HCl \quad \rightarrow \quad Cl_2 + H_2O \tag{R2}$$
$$Cl_2 + h\nu \quad \rightarrow \quad 2\,Cl \tag{R8}$$
$$Cl + O_3 \quad \rightarrow \quad ClO + O_2 \quad (2\times) \tag{R9}$$
$$Net(C1): \quad CH_4 + 2\,O_3 \quad \rightarrow \quad CH_2O + H_2O + 2\,O_2 \tag{R10}$$

Cycle C1 was first formulated by Crutzen et al. (1992) demonstrating the importance of reaction R3 for ozone hole chemistry. Reaction R3 is essential for the production of a $HO_2$ radical from the $CH_3$ formed in reaction R4.

$$CH_2O + Cl \quad \rightarrow \quad HCl + CHO \tag{R11}$$
$$CHO + O_2 \quad \rightarrow \quad CO + HO_2 \tag{R12}$$
$$ClO + HO_2 \quad \rightarrow \quad HOCl + O_2 \tag{R7}$$
$$HOCl + HCl \quad \rightarrow \quad Cl_2 + H_2O \tag{R2}$$
$$Cl_2 + h\nu \quad \rightarrow \quad 2\,Cl \tag{R8}$$
$$Cl + O_3 \quad \rightarrow \quad ClO + O_2 \tag{R9}$$
$$Net(C2): \quad CH_2O + O_3 \quad \rightarrow \quad CO + H_2O + O_2 \tag{R13}$$

In cycles C1 and C2, HOCl is produced at the same rate as HCl because in both reactions R4 and R11, for each HCl molecule produced, also an $HO_2$ radical is generated (from $CH_3$ via R5, R3, and R6 or from CHO via R12). This $HO_2$ radical reacts further to form HOCl, which then reacts heterogeneously with HCl (in reaction R2) so that there is no net production of HCl. We therefore refer to cycles C1 and C2 in the following as "HCl null-cycles".





For the HCl null-cycle argument it is essential that Reaction 3 dominates the loss of $CH_3O_2$ and indeed alternative reactions for $CH_3O_2$ (e.g. the $CH_3O_2$ selfreaction) were discussed (Müller and Crutzen, 1994). However, all relevant alternative reactions are included in the chemical scheme used here and the competition of the different reaction pathways of $CH_3O_2$ is addressed in the pathway analysis that was employed (Lehmann, 2004).

5      These HCl null-cycles are effective in ensuring that there is no net production of HCl even though the speed of reactions R4 and R11, and thus the production of HCl, increases by about two orders of magnitude during September (Figure 1, panel g).

     We find that during the first 10 days of September, HCl is recycled at a slow rate of 88 ppt in 10 days through cycle C1, at a rate of 26 ppt in 10 days through cycle C2. However, for the last days of September (21-30), the rate of recycling of HCl in cycles C1 and C2 is much more rapid, the rates are 3.7 ppb in 10 days and 2.9 ppb in 10 days respectively.

10      Thus cycles C1 and C2 constitute the chemical mechanism responsible for the maintenance of high levels of active chlorine (and thus of continued ozone destruction) under conditions of increasingly rapid HCl formation in reactions R4 and R11.

### 3.2    The path to full activation of HCl

However cycles C1 and C2 cannot explain the decrease of HCl from values of $\approx 1$ ppb in early July to very low values (about 0.2 ppb) during late August and September and thus the complete activation of chlorine as observed (Santee et al., 2008).

     Using the pathway analysis (Lehmann, 2004), we identify two chemical cycles, C3 and C4 which are responsible for the decline of HCl during August and September:

$$CH_2O + h\nu \quad \rightarrow \quad CHO + H \tag{R14}$$

$$H + O_2 \quad \rightarrow \quad HO_2 + M \tag{R15}$$

$$CHO + O_2 \quad \rightarrow \quad CO + HO_2 \tag{R12}$$

$$ClO + HO_2 \quad \rightarrow \quad HOCl + O_2 \quad (2\times) \tag{R7}$$

$$HOCl + HCl \quad \rightarrow \quad Cl_2 + H_2O \quad (2\times) \tag{R2}$$

$$Cl_2 + h\nu \quad \rightarrow \quad 2\,Cl \quad (2\times) \tag{R8}$$

$$Cl + O_3 \quad \rightarrow \quad ClO + O_2 \quad (4\times) \tag{R9}$$

$$Net(C3): \quad CH_2O + 2\,HCl + 4\,O_3 \quad \rightarrow \quad CO + 2\,ClO + 2\,H_2O + 4\,O_2 \tag{R16}$$



$$O_3 + h\nu \quad \rightarrow \quad O(^1D) + O_2 \tag{R17}$$

$$O(^1D) + H_2O \quad \rightarrow \quad 2\,OH \tag{R18}$$

$$OH + O_3 \quad \rightarrow \quad HO_2 + O_2 \quad (2\times) \tag{R19}$$

$$ClO + HO_2 \quad \rightarrow \quad HOCl + O_2 \quad (2\times) \tag{R7}$$

$$HOCl + HCl \quad \rightarrow \quad Cl_2 + H_2O \quad (2\times) \tag{R2}$$

$$Cl_2 + h\nu \quad \rightarrow \quad 2\,Cl \quad (2\times) \tag{R8}$$

$$Cl + O_3 \quad \rightarrow \quad ClO + O_2 \quad (4\times) \tag{R9}$$

$$Net(C4): \quad 2\,HCl + 7\,O_3 \quad \rightarrow \quad 2\,ClO + H_2O + 9\,O_2 \tag{R20}$$

Reaction R19 may also proceed in two steps:

$$OH + ClO \quad \rightarrow \quad HO_2 + Cl \tag{R21}$$

$$Cl + O_3 \quad \rightarrow \quad ClO + O_2 \tag{R9}$$

$$Net: \quad OH + O_3 \quad \rightarrow \quad HO_2 + O_2 \tag{R22}$$

First, both cycles C3 and C4 require sufficiently fast heterogeneous reactions to be present. Second, in both cycles, it is important that $HO_2$ radicals are produced *without* simultaneous HCl formation (in contrast to cycles C1 and C2). The $HO_2$ radicals lead to formation of HOCl so that there is net HCl loss through reaction R2. The formation of $HO_2$ radicals in C3 and C4 is thus the key process responsible for the decline of HCl in August and September.

The dominant source of $HO_x$ radicals under the conditions of the polar lower stratosphere in late winter and early spring is not the production of $O(^1D)$ radicals through ozone photolysis with subsequent reaction with $H_2O$, but rather the the radical channel of the photolysis of $CH_2O$ (reaction R14) (Crutzen et al., 1992; Müller and Crutzen, 1994; Crowley et al., 1994).

Thus, the photolysis of $CH_2O$ (radical channel) is effectively driving the depletion of HCl. In the timeframe 20-31 August, 66 % of the net HCl depletion occurs through the photolysis of $CH_2O$ and cycle C3 and only 15 % through the formation of $O(^1D)$ and subsequent reaction with $H_2O$ (cycle C4). The rate of production of $ClONO_2$, under the conditions considered here is very low (see discussion below).

The photolysis of $CH_2O$ possesses two product channels; reaction R14 (radical channel) leading to the formation of two $HO_2$ radicals and the molecular channel

$$CH_2O + h\nu \rightarrow CO + H_2 \tag{R23}$$

which does not lead to production of $HO_x$. The branching ratio between reactions R14 and R23 is uncertain; it is about 30% for the radical channel R14 for the conditions in question here (Röth and Ehhalt, 2015). Furthermore, both the photolysis cross sections and the branching ratio for the photolysis of $CH_2O$ are likely temperature dependent (with the intensities of the



**Figure 2.** Sensitivity of the ozone hole chemistry on $HO_2$ production in the photolysis of $CH_2O$. Black lines show the reference case, red line a case assuming 100% efficiency for $HO_2$ production (i.e. for reaction R14), and blue line a case assuming no $HO_2$ production in the photolysis of $CH_2O$ (100% efficiency for reaction R23).



maxima of each absorption band increasing with lower temperature), but this dependence is not well characterised in laboratory measurements for temperatures below 200 K (Smith et al., 2006; Röth and Ehhalt, 2015) which are relevant here.

To demonstrate the importance of cycle C3 and in particular the $HO_2$ production through reaction R14, we conducted two sensitivity runs, one assuming 100% efficiency for radical channel R14 and zero for the molecular channel R23 (red line in

Figure 2) and vice versa (blue line in Figure 2). These assumptions constitute upper and lower limits of $HO_2$ production in reaction R14 that will not be reached in reality (see methods).

Without $HO_2$ production in the $CH_2O$ photolysis there is little reduction in HCl between late August and late September and, consequently, lower $ClO_x$ prevails leading to slower ozone loss than in the reference case. Assuming 100% efficiency for the $HO_2$ producing channel in the $CH_2O$ photolysis (red line in Figure 2) results in a much more rapid depletion of HCl than

in the reference case and near zero values of HCl are reached in late August. Consequently, in this case, values of $ClO_x$ are higher and ozone destruction is faster. Minimum values of ozone are reached somewhat earlier and likewise the corresponding rapid increase (deactivation) of HCl occurs somewhat earlier.

During the period of near zero values of HCl during late August to mid September, no substantial decrease of $ClO_x$ and thus no deactivation through an increase in HCl occurs (red line in Figure 2). This is the case because cycles C1 and C2 (net

reactions R10 and R13) are still active and prevent the net formation of HCl. Solely the ongoing production of $HO_2$ radicals causes somewhat enhanced levels of HOCl (0.1 to 0.2 ppb) during this period. Photolysis of HOCl inhibits an accumulation of larger amounts of HOCl.

### 3.3   The role of $ClONO_2$ and denitrification

For the conditions in the heart of the ozone layer, in the lower stratosphere, which are considered here, reaction

$$ClO + NO_2 + M \rightarrow ClONO_2 + M \tag{R24}$$

and the related chemistry involving $ClONO_2$ only play a minor role. This is the case mainly because polar stratospheric clouds (PSCs) consisting of NAT or STS (Supercooled ternary solutions) exist almost continuously throughout the simulation sequestering most $HNO_3$ from the gas-phase. This leads to low $NO_2$ concentrations even in the presence of sunlight in spring. Consequently, throughout Antarctic winter and spring, and in particular during the main ozone loss period from late August to

late September, the rate of the heterogeneous reaction between HCl and HOCl (R2) is substantially larger than the rate of HCl + $ClONO_2$ (R1), which is caused by the slow rate of formation of $ClONO_2$. The ClO radical preferentially reacts with $HO_2$ (forming HOCl) rather than with $NO_2$ (forming $ClONO_2$).

For the efficacy of cycles C1 and C2, low gas-phase concentrations of $NO_2$ (and thus of $HNO_3$) are necessary. It is however not important whether the removal of $HNO_3$ from the gas-phase occurs temporarily (uptake in particles) or permanently (deni-

trification). In our box-model simulations the impact of denitrification is taken into account by assuming that denitrification has occurred by the start of the simulation (4.5 ppb initial $HNO_3$ at 1 June). Overall, for the conditions in the core of the Antarctic vortex considered here, there is very little impact of denitrification on ozone depletion (see appendix).



### 3.4 Sensitivity of ozone loss on stratospheric methane and chlorine levels

In the coming decades, the composition of the Antarctic lower stratosphere will change considerably, in particular the stratospheric halogen loading will continue to decrease (e.g., WMO, 2014; Chipperfield et al., 2015). Repeating the reference run with $Cl_y$ halved (blue line in Figure 3) (typical conditions for $\approx$2050) results in lower active chlorine $ClO_x$ and thus slower

ozone loss rates, as expected. However, in this case the ozone depletion period in the lower stratosphere is longer, so that extremely low ozone values (below 0.2 ppb) are reached also for the initial amount of $Cl_y$ halved, albeit about three weeks later than in the reference case. The fact that in our simulation the impact of a reduced amount of $Cl_y$ is mainly observed in September is consistent with the recent conclusion that signs of healing in the Antarctic ozone layer have emerged for September (Solomon et al., 2016). Note that in the reference case ozone depletion stops because ozone is reaching extremely low values

although temperatures are still low enough for PSCs and heterogeneous processing; the same mechanism works for halved $Cl_y$, solely the low values are reached later in the season due to the slower ozone loss rate.

Doubling methane in the simulation leads to a speed up of Reaction R4, thereby also to enhanced production of $CH_2O$ and thus also to a speed up of Reaction R11. But doubling methane does not lead to an enhanced deactivation, because cycles C1 and C2 are active and inhibit the deactivation effect of Reactions R4 and R11. However, surprisingly at first, doubling of

methane means that more $CH_2O$ is produced in cycle C1 leading to somewhat faster HCl activation in late August and early September through cycle C3 and thus to higher $ClO_x$ and a somewhat faster ozone depletion during this period.

A simulation combining doubled methane and halved $Cl_y$ shows results very close to those for halved $Cl_y$, but again the doubled methane does not lead to a substantially faster deactivation and practically the same very low ozone values are reached in late October. Thus we suggest that the very low ozone values observed today in the core of the polar vortex (Solomon et al.,

2005) will continue to be measured for decades to come and that the recovery of the stratospheric chlorine loading should lead to a shift by some weeks of the low values to later in the season, consistent with recent observations (Solomon et al., 2016).

### 3.5 Multi-trajectory simulations

We have repeated the reference run for a set of realistic trajectories in the vicinity of the South pole including diabatic descent and latitude variations (Grooß et al., 2011). The initial values (for 1 August) of HCl and $ClO_x$ were chosen consistently with

the reference run. The results of these runs (Fig. 4) show a significant variability in the decline of HCl and ozone loss rate, but all show a rather similar behaviour as the reference run. Namely, very little chemical change in polar night until early August ("sleeping chemistry") and a relatively slow additional chlorine activation until early September. Next, a maintenance of high $ClO_x$ values during mid-September to early October (due to cycles C1 and C2) accompanied by rapid ozone loss. This period of high $ClO_x$ ends abruptly with the rapid formation of HCl (and thus chlorine deactivation), which occurs in a similar way for

the individual trajectories in the time frame late September to early October.







**Figure 3.** Sensitivity of the ozone hole chemistry on stratospheric methane and chlorine levels. Black lines show the reference case, red line a case assuming initial methane to be doubled, blue line a case assuming initial inorganic chlorine ($Cl_y$) to be halved, and green line a case for initial methane doubled and $Cl_y$ halved.





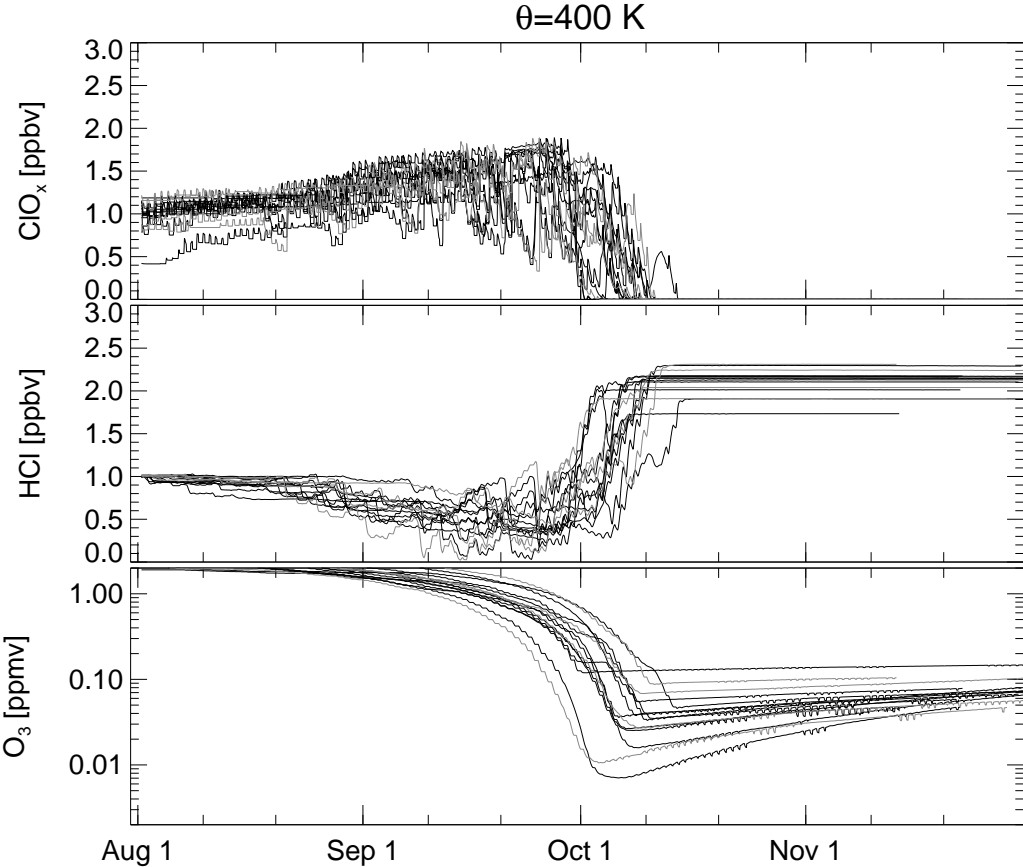

**Figure 4.** Results from multi-trajectory simulations of CLaMS. Box-model simulations were performed for a set of trajectories passing the South pole at 400 K taken from (Grooß et al., 2011). The box-model simulations cover the time period from 1 August to 30 November 2003. A few trajectories showing very little diabatic descent (and thus much smaller values of total chlorine) were neglected. Initial values were chosen consistently with the reference run (Fig. 1). Individual trajectories are shown in different shades of grey to allow them to be distinguished more easily.





## 4  Conclusions

For the heart of the ozone hole in the Antarctic lowermost stratosphere, formation of HCl through reactions R4 and R11 is very rapid for enhanced chlorine levels with the rate of HCl formation increasing by more than a factor of ten during September. We have shown that high levels of active chlorine are maintained nonetheless because formation of HCl is balanced in very effective

HCl null-cycles, allowing rapid chemical destruction of ozone to proceed. Further, for the depletion of HCl to very low values, formation of $HO_2$ is essential, with photolysis of $CH_2O$ being the major net source of $HO_2$. Owing to the uptake of $HNO_3$ in NAT and STS particles, $NO_x$ chemistry and the formation of $ClONO_2$ are of minor importance. Chlorine is finally deactivated if an imbalance in the null-cycles occurs leading to a rapid and almost complete conversion of the activated chlorine into HCl and thereby putting a halt to ozone depletion (Douglass et al., 1995; Grooß et al., 1997; Grooß et al., 2011). An increase of

methane in the future should not lead to a faster ozone recovery as might be expected. These results are important for an assessment of the impact of chemical change to come in the Antarctic stratosphere on the future development of the ozone hole.

*Data availability.* The model results presented here can be obtained in electronic form (netcdf-files) from the corresponding author on request.

*Author contributions.* R.M., J.-U.G. and R.L. conceived and designed the research project. A.M.Z. and J.-U.G. conducted the simulations. R.L. performed the pathway analysis. R.M., J.-U.G., M.A.Z. and R.L. contributed to the interpretation of the results and wrote the paper

*Competing interests.* The authors declare that they have no competing financial interests

*Acknowledgements.* We thank Sabine Robrecht for helpful discussions. Part of this work was funded by the European Community's Seventh Framework Programme (FP7/2007–2013) as part of the StratoClim project (grant agreement no. 603557). We thank the European Centre for

Medium-range Weather Forecasts (ECMWF) for providing meteorological data sets.



## Appendix A: Sensitivity studies

In this section we discuss the sensitivity of the simulated development of chlorine chemistry and ozone mixing ratios with respect to the initial ozone and HNO₃ mixing ratios Grooß et al. (similar as in 2011).

### A1 Initial ozone mixing ratio

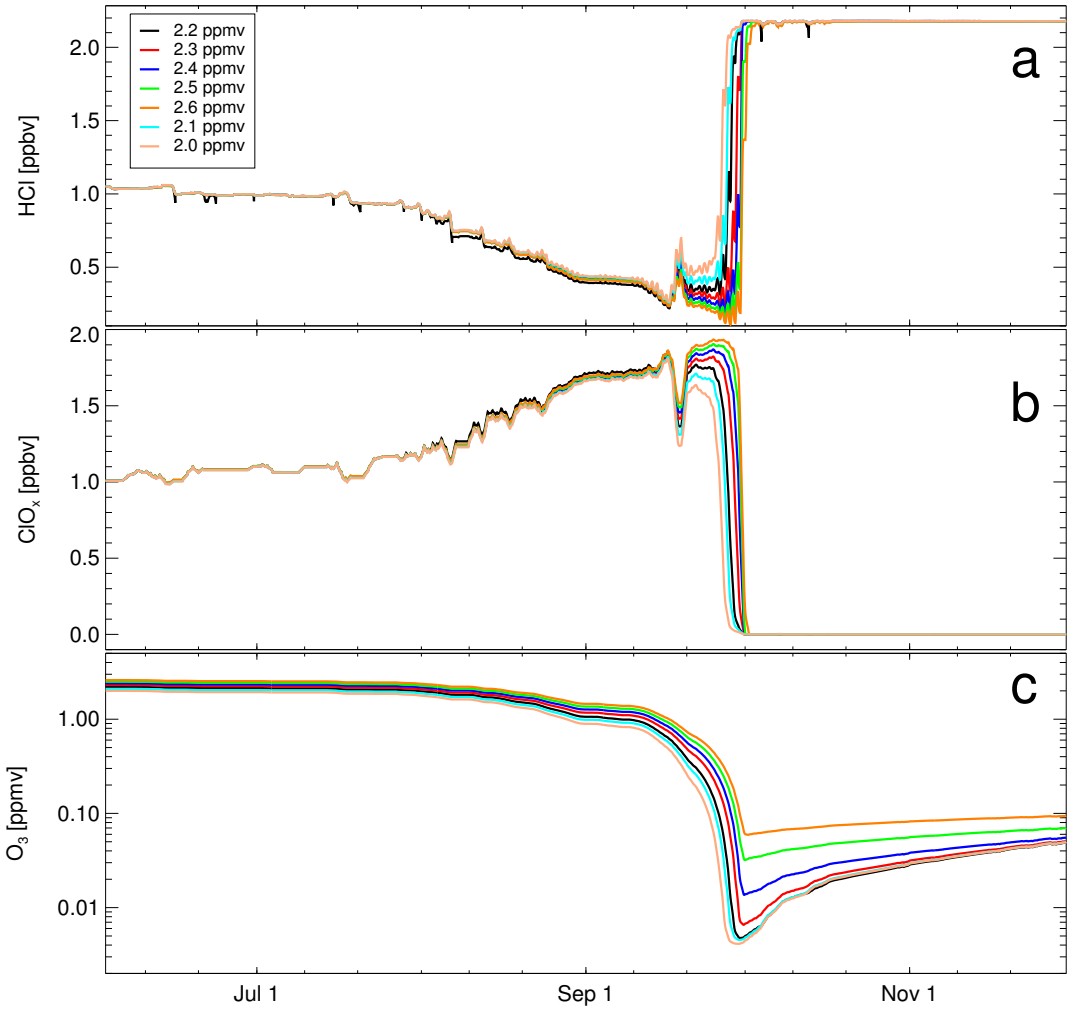

**Figure A1.** Simulations for different initial ozone mixing ratios. Simulation as in the reference run but with different initial ozone mixing ratios (2.0 to 2.6 ppm).

5      We have conducted six additional box-model simulations, identical to the reference run (initial ozone 2.2 ppm), but with different initial ozone mixing ratios ranging from 2.0 to 2.6 ppm (Fig. A1). The simulations show both different minimum ozone mixing ratios and different times at which these minima are reached (consistent with the results of Grooß et al., 2011).




Similarly, the timing of the rapid increase in HCl and chlorine deactivation is different for the different initial ozone mixing ratios (Fig. A1).

Importantly however, the general pattern of chlorine activation and decrease of HCl is very similar until about 20 September indicating that the efficacy of the HCl null-cycles is not affected by the initial ozone mixing ratio.

## 5  A2  Denitrification

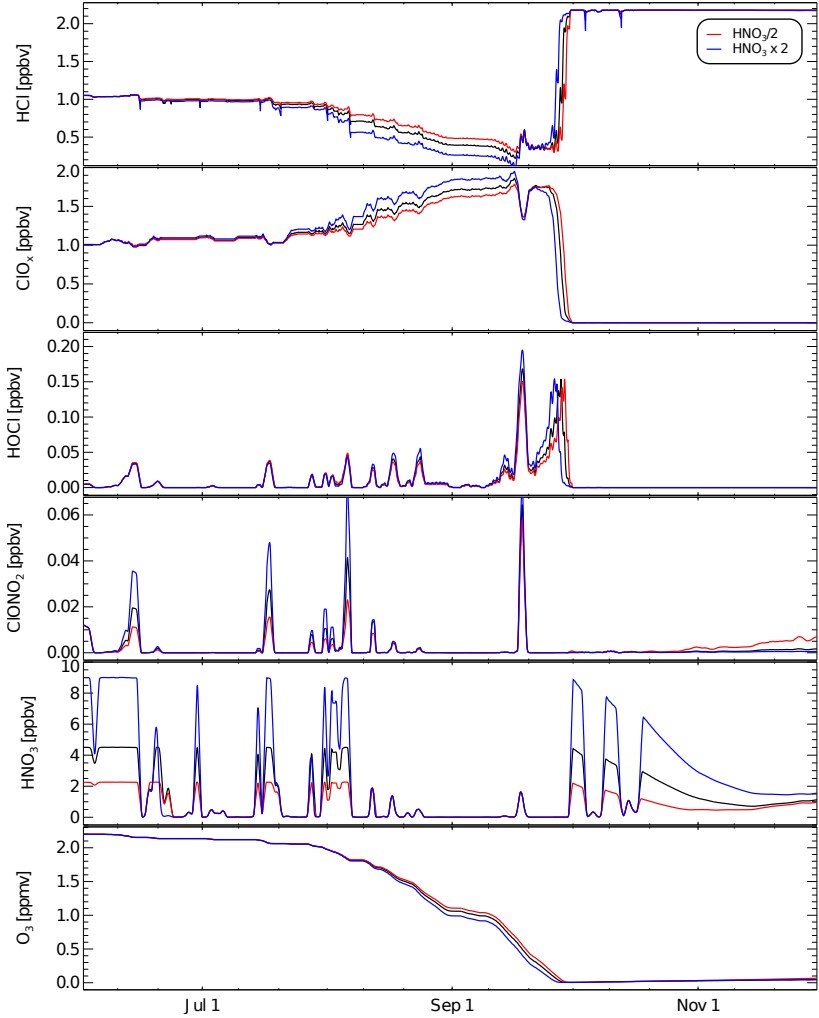

**Figure A2.** Simulation for the impact of denitrification. The impact of denitrification is tested by initialising the model run with HNO$_3$ doubled and halved.

The formation of polar stratospheric clouds impacts polar chemistry by taking up HNO$_3$ and thereby reducing HNO$_3$ in the gas-phase (Crutzen and Arnold, 1986; Toon et al., 1986; Salawitch et al., 1988). In the case of supercooled ternary solution





(STS) particles the reduction of gas-phase $HNO_3$ lasts only as long as temperatures are low enough for the particles to exist, because the particles are too small to sediment significantly. Nitric acid trihydrate particles, however, can grow to large sizes allowing substantial sedimentation speeds, so that $HNO_3$ is removed from stratospheric air masses permanently (a process referred to as "denitrification", Fahey et al., 2001; Molleker et al., 2014). Denitrification is observed regularly in the lower
stratosphere in Antarctica and in cold winters in the Arctic (e.g., Davies et al., 2006; Santee et al., 2008).

For the case considered here, PSCs that sequester $HNO_3$ from the gas-phase exist almost continuously until mid September (Fig. 1). Therefore, the gas-phase concentration of $HNO_3$ during the period of chlorine activation, maintenance of active chlorine, and chemical ozone depletion is controlled by PSCs and not by denitrification.

Nonetheless, we conducted a sensitivity simulation initialised with half (2.25 ppb) or twice (9 ppb) the amount of initial
$HNO_3$ (Fig. A2). These assumptions result in practically the same ozone depletion in late September as in the reference case (Fig. A2). For mid September, the simulation with 9 ppb initial $HNO_3$ shows about 10% more active chlorine than the reference run and thus a somewhat more rapid ozone loss (ozone in mid September $\approx 0.1$ ppm less than in the reference run). For half the initial $HNO_3$ (2.25 ppb) the situation is reversed, but differences are even smaller.

Thus, as to be expected, the impact of the initial $HNO_3$ mixing ratio on the simulated ozone loss and chlorine activation
is only moderate. Solely in the short periods when $HNO_3$ is released to the gas-phase and some of this gas-phase $HNO_3$ is converted to $NO_x$, the resulting production of $ClONO_2$ is enhanced in the case of a greater initial $HNO_3$ mixing ratio. Therefore, a greater initial $HNO_3$ mixing ratio (assumption of no denitrification) leads to a somewhat faster HCl depletion (through R1), a stronger chlorine activation and thus a little bit more chemical ozone loss (Fig. A2).

Overall, like for the sensitivity on the initial ozone mixing ratio, the general pattern of chlorine activation and decrease of
HCl is very similar for the results of the reference run and the results under the (rather extreme) assumptions of reduced and enhanced denitrification (Fig. A2).



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
