# Peer review of "The maintenance of elevated active chlorine levels in the Antarctic lower stratosphere through HCl null-cycles"

_Atmospheric Chemistry and Physics, 2017_

## Referee Comment (RC1) · Anonymous Referee #2 · 16 Nov 2017

**1   General remarks**

The paper presents a classical process study on important details of the Antarctic ozone hole and is suitable for ACP after minor improvement.

[Figure]

**2 Specific remarks**

Page 3, line 28 and Figure 2: Wouldn't it be better to use for the range of HCHO photolysis branching cases near the experimental uncertainty? The totally unrealistic limits give a larger separation of the curves but what do we learn from that?

Page 7 and 8, cycles C3 and C4: These cycles require a lot of ozone. Late September or early October ozone might be too depleted for that and chlorine deactivation starts. Please give some remarks on that, best with typical threshold values. It is too difficult to estimate that from the figures like presented.

Section 3.5: There CLAMS should be mentioned (cited) also in the text and not only in the caption of Fig.4 and in section 2.1.

**3 Technical corrections**

Figure 4 might be easier to read with colored curves for the extreme cases.

Page 14, line 16: Initials messed up?

Page 15, line 3: order of words or parentheses wrong.

Figure A1: A step of 0.2ppmv and an ordered legend would be better.

---

## Referee Comment (RC2) · Anonymous Referee #1 · 21 Nov 2017

The manuscript by Müller et al. represents an important process study for stratospheric chemistry in the Antarctic. The manuscript details a mechanism for the maintenance of high ClOx through effective chemical cycles (termed HCl null-cycles) that inhibit chlorine deactivation. The authors apply state-of-the-art box model calculations to determine chemical reaction rates and chemical pathways, explore the effects of future changes in chlorine and methane levels and provide a sensitivity analysis for different initial ozone mixing ratios and HNO3 levels.

The manuscript is well prepared, and I find it suitable for publication in ACP after a few minor additions/corrections detailed below.

Specific Comments:

P3, L13: please provide briefly a few specifics of the radiation code applied

P3, L32: I understand why the authors use the 0% and 100% branching ratios as limit cases, but performing integrations with a small set of intermediate, more realistic, branching ratios would strengthen the manuscript.

P11, L24: please provide a little more detail on the diabatic descent and latitude variations considered

Technical Comments:

Figures 2, 3, A1, A2: readability of panel (a) would be improved by extending the axis to 0 ppb.

Figure 4: adding the results of the reference simulation (in color) to this figure would be helpful

All Figures: alternating the labels between left and right y-axis among panels would improve readability and allow for larger axis labels.

---

## Author Comment (AC1) · 8 Dec 2017

We thank the referee for the review and for very helpful comments. We give a point-by-point reply below, where the reviewer comments are repeated in italics.

General

*The manuscript by Müller et al. represents an important process study for stratospheric chemistry in the Antarctic. The manuscript details a mechanism for the maintenance of high ClOx through effective chemical cycles (termed HCl null-cycles) that inhibit*

*chlorine deactivation. The authors apply state-of-the-art box model calculations to determine chemical reaction rates and chemical pathways, explore the effects of future changes in chlorine and methane levels and provide a sensitivity analysis for different initial ozone mixing ratios and HNO3 levels. The manuscript is well prepared, and I find it suitable for publication in ACP after a few minor additions/corrections detailed below.*

Thank you very much. In the revised version all comments have been taken into account.

Specific Comments

*P3, L13: please provide briefly a few specifics of the radiation code applied*

We agree, we have added the following text to the paper: "The diabatic descent rates were calculated using a radiation code (Morcrette, 1991; Zhong and Haigh, 1995) assuming a cloud-free atmosphere. We use temperatures from the ECMWF operational analyses and climatological ozone and water vapour profiles (Grooß and Russell, 2005)."

*P3, L32: I understand why the authors use the 0% and 100% branching ratios as limit cases, but performing integrations with a small set of intermediate, more realistic, branching ratios would strengthen the manuscript.*

We agree and have conducted additional simulations assuming a 20% increase (yellow line in Fig. 1) and decrease (light blue line in Fig. 1) of the recommended branching ratio for the radical channel of the formaldehyde photolysis; the value of 20% is deduced from the study of Röth and Ehhalt (2015). The results are shown in Fig. 1 and will be added to the paper.

*P11, L24: please provide a little more detail on the diabatic descent and latitude varia-*

*tions considered*

We agree. We have inserted the following text in the paper: "In the period early August to early October all trajectories are subject to roughly the same diabatic descent of $\approx 10\,\mathrm{K}$, similarly as for the reference run. In this period, all trajectories show strong variations in latitude, again similar as for the reference run. The latitude varies between the South Pole and $\approx 65°$S with some equatorward excursions to $\approx 60°$S or, even more rarely, to $\approx 55°$S."

Technical Comments

*Figures 2, 3, A1, A2: readability of panel (a) would be improved by extending the axis to 0 ppb.*

Thanks very much for pointing out this oversight! All figures are corrected in the revised version.

*Figure 4: adding the results of the reference simulation (in color) to this figure would be helpful*

We agree. The revised figure (Fig. 2 of this reply) is shown here.

*All Figures: alternating the labels between left and right y-axis among panels would improve readability and allow for larger axis labels.*

As suggested, we have increased the size of the axis labels.

**References**

Grooß, J.-U. and Russell, J. M.: Technical note: A stratospheric climatology for $O_3$, $H_2O$, $CH_4$,

NO$_x$, HCl, and HF derived from HALOE measurements, Atmos. Chem. Phys., 5, 2797–2807, 2005.

Grooß, J.-U., Brautzsch, K., Pommrich, R., Solomon, S., and Müller, R.: Stratospheric ozone chemistry in the Antarctic: What controls the lowest values that can be reached and their recovery?, Atmos. Chem. Phys., 11, 12 217–12 226, 2011.

Morcrette, J.-J.: Radiation and cloud radiative properties in the European Centre for Medium-Range Weather Forecasts forecasting system, J. Geophys. Res., 96, 9121–9132, 1991.

Röth, E.-P. and Ehhalt, D. H.: A simple formulation of the CH$_2$O photolysis quantum yields, Atmos. Chem. Phys., 15, 7195–7202, doi:10.5194/acp-15-7195-2015, http://www.atmos-chem-phys.net/15/7195/2015/, 2015.

Zhong, W. and Haigh, J. D.: Improved broadband emissivity parameterization for water vapor cooling rate calculations, J. Atmos. Sci., 52, 124–138, 1995.
* * *
[Figure]

**Fig. 1.** Improved version of Fig.∼2 showing the effect of realistic estimates of the uncertainty of the branching ratio of the formaldehyde photolysis (light blue and yellow lines).

θ=400 K

**Fig. 2.** Results from multi-trajectory simulations of CLaMS. Box-model simulations were performed for a set of trajectories passing the South pole at 400 taken from Grooß, et al. 2011.

---

## Author Comment (AC2) · 8 Dec 2017

We thank the referee for the review and for very helpful comments. We give a point-by-point reply below, where the reviewer comments are repeated in italics.

General remarks

*The paper presents a classical process study on important details of the Antarctic ozone hole and is suitable for ACP after minor improvement.*

Thank you very much. In the revised version all comments have been taken into ac-

count.

Specific remarks

*Page 3, line 28 and Figure 2: Wouldn't it be better to use for the range of HCHO photolysis branching cases near the experimental uncertainty? The totally unrealistic limits give a larger separation of the curves but what do we learn from that?*

We agree and have conducted additional simulations assuming a 20% increase (yellow line in Fig. 1 of this reply) and decrease (light blue line in Fig. 1) of the recommended branching ratio for the radical channel of the formaldehyde photolysis; the value of 20% is deduced from the study of Röth and Ehhalt (2015). The results are shown in Fig. 1 of this reply and will be added to the paper.

However, assuming no $HO_2$ production in the photolysis of $CH_2O$ (blue line in Fig. 1 leads to very little reduction in HCl between late August and late September. This corroborates the conclusion that the HCl depletion beyond the initial titration against $ClONO_2$ is driven by the radical channel of the $CH_2O$ photolysis. In line with the recommendation of reviewer #1, we have therefore retained the dark blue and red line in the figure.

*Page 7 and 8, cycles C3 and C4: These cycles require a lot of ozone. Late September or early October ozone might be too depleted for that and chlorine deactivation starts. Please give some remarks on that, best with typical threshold values. It is too difficult to estimate that from the figures like presented.*

The reviewer is correct in pointing out that the net reactions for cycle C3 and C4 require the presence of ozone (three and seven ozone molecules in cycles C3 and C4 respectively). These molecules are required for reaction R9:

$$Cl + O_3 \rightarrow ClO + O_2 \tag{1}$$

reaction R17:

$$O_3 + h\nu \rightarrow O(^1D) + O_2 \qquad (2)$$

and reaction R19:

$$OH + O_3 \rightarrow HO_2 + O_2; \qquad (3)$$

these reactions are fast enough to sustain cycles C3 and C4 even at low ozone concentrations. However, we agree that it is necessary to include some discussion on the issue to the paper. We have now inserted the following sentences in the text after cycles C3 and C4 are introduced:

"Reactions R9, R17, and R19 are fast enough to sustain cycles C3 and C4 even at the very low ozone concentrations as they occur in late September. For example, in the reference run for 26 September, the diurnal mean ozone mixing ratio is 77 ppb, the rate of reaction R9 is 46 ppb/day, the rate of reaction R17 is 15 ppb/day, and the net rate of reaction R19 is $3.2 \cdot 10^{-2}$ ppb/day directly and 1.1 ppp/day, when it proceeds in two steps (R21 and R9). Therefore, even at extremely low ozone concentrations in late September, the rates of reactions R9, R17, and R19 are not rate limiting for cycles C3 and C4. Under these conditions, the rate limiting reaction for cycle C3 is the radical channel of the photolysis of $CH_2O$ and for cycle C4 the reaction of $O(^1D)$ with water vapour. (Note that reactions R9 and R17 do not constitute the rate limiting step of ozone loss cycles and can therefore not be used to deduce the ozone loss rate)."

*Section 3.5: There CLAMS should be mentioned (cited) also in the text and not only in the caption of Fig.4 and in section 2.1.*

We agree, the first sentences of section 3.5 read now: "We have repeated the reference run using the CLaMS model (McKenna et al., 2002; Grooß et al., 2005) in box-model mode for multiple trajectories. We employ a set of realistic trajectories passing the South pole at 400 K potential temperature (in late September/early October) including diabatic descent and latitude variations (taken from Grooß et al., 2011)." . . .
[Figure]

Technical corrections

*Figure 4 might be easier to read with colored curves for the extreme cases.* This would be one option to improve Fig. 4; here we have followed the suggestion from reviewer #1 to emphasise the results from the reference run in colour.

*Page 14, line 16: Initials messed up?* Yes, it should read "A.M.Z." – corrected.

*Page 15, line 3: order of words or parentheses wrong.* Yes, citation is corrected.

*Figure A1: A step of 0.2ppmv and an ordered legend would be better.* We agree. The revised version of the plot shows a step of 0.2 ppmv and an ordered legend as suggested.

**References**

Grooß, J.-U., Günther, G., Müller, R., Konopka, P., Bausch, S., Schlager, H., Voigt, C., Volk, C. M., and Toon, G. C.: Simulation of denitrification and ozone loss for the Arctic winter 2002/2003, Atmos. Chem. Phys., 5, 1437–1448, 2005.

Grooß, J.-U., Brautzsch, K., Pommrich, R., Solomon, S., and Müller, R.: Stratospheric ozone chemistry in the Antarctic: What controls the lowest values that can be reached and their recovery?, Atmos. Chem. Phys., 11, 12 217–12 226, 2011.

McKenna, D. S., Grooß, J.-U., Günther, G., Konopka, P., Müller, R., Carver, G., and Sasano, Y.: A new Chemical Lagrangian Model of the Stratosphere (CLaMS): 2. Formulation of chemistry scheme and initialization, J. Geophys. Res., 107, 4256, doi:10.1029/2000JD000113, 2002.

Röth, E.-P. and Ehhalt, D. H.: A simple formulation of the $CH_2O$ photolysis quantum yields, Atmos. Chem. Phys., 15, 7195–7202, doi:10.5194/acp-15-7195-2015, http://www.atmos-chem-phys.net/15/7195/2015/, 2015.

[Figure]

Figure with four stacked panels showing time series from around Jul 1 to Nov 1.

Panel 1 (y-axis: HCl [ppbv], range 0.0–2.0) legend:
- CH$_2$O+h$\nu$→CHO+H (red)
- CH$_2$O+h$\nu$→CO+H$_2$ (blue)
- CH$_2$O+h$\nu$→CHO+H (enh.) (orange)
- CH$_2$O+h$\nu$→CHO+H (red.) (cyan)

Panel 2 (y-axis: HOCl [ppbv], range 0.00–0.30)

Panel 3 (y-axis: ClO$_x$ [ppbv], range 0.0–2.0)

Panel 4 (y-axis: O$_3$ [ppmv], log scale 0.01–1.00)

**Fig. 1.** Improved version of Fig.~2 showing the effect of realistic estimates of the uncertainty of the branching ratio of the formaldehyde photolysis (light blue and yellow lines).